# Sustained Systemic Antioxidative Effects of Intermittent Theta Burst Stimulation beyond Neurodegeneration: Implications in Therapy in 6-Hydroxydopamine Model of Parkinson’s Disease

**DOI:** 10.3390/antiox13020218

**Published:** 2024-02-08

**Authors:** Milica Zeljkovic Jovanovic, Jelena Stanojevic, Ivana Stevanovic, Milica Ninkovic, Nadezda Nedeljkovic, Milorad Dragic

**Affiliations:** 1Laboratory for Neurobiology, Department for General Physiology and Biophysics, Faculty of Biology, University of Belgrade, 11000 Belgrade, Serbia; nnedel@bio.bg.ac.rs; 2Institute for Medical Research, Military Medical Academy, 11000 Belgrade, Serbia; jelena.stanojevic@mod.gov.rs (J.S.); ivana.stevanovic@vma.mod.gov.rs (I.S.); milica.ninkovic@vma.mod.gov.rs (M.N.); 3Medical Faculty of Military Medical Academy, University of Defense, 11000 Belgrade, Serbia; 4Department of Molecular Biology and Endocrinology, VINČA Institute of Nuclear Sciences-National Institute of the Republic of Serbia, University of Belgrade, 11351 Belgrade, Serbia

**Keywords:** Parkinson’s disease, 6-hydroxidopamine, rTMS, intermittent theta burst stimulation, oxidative stress, neuroprotection

## Abstract

Parkinson’s disease (PD) is manifested by the progressive loss of dopaminergic neurons in the substantia nigra pars compacta (SNpc) and caudoputamen (Cp), leading to the development of motor and non-motor symptoms. The contribution of oxidative stress to the development and progression of PD is increasingly recognized. Experimental models show that strengthening antioxidant defenses and reducing pro-oxidant status may have beneficial effects on disease progression. In this study, the neuroprotective potential of intermittent theta burst stimulation (iTBS) is investigated in a 6-hydroxydopamine (6-OHDA)-induced PD model in rats seven days after intoxication which corresponds to the occurrence of first motor symptoms. Two-month-old male Wistar rats were unilaterally injected with 6-OHDA to mimic PD pathology and were subsequently divided into two groups to receive either iTBS or sham stimulation for 21 days. The main oxidative parameters were analyzed in the caudoputamen, substantia nigra pars compacta, and serum. iTBS treatment notably mitigated oxidative stress indicators, simultaneously increasing antioxidative parameters in the caudoputamen and substantia nigra pars compacta well after 6-OHDA-induced neurodegeneration process was over. Serum analysis confirmed the systemic effect of iTBS with a decrease in oxidative markers and an increase in antioxidants. Prolonged iTBS exerts a modulatory effect on oxidative/antioxidant parameters in the 6-OHDA-induced PD model, suggesting a potential neuroprotective benefit, even though at this specific time point 6-OHDA-induced oxidative status was unaltered. These results emphasize the need to further explore the mechanisms of iTBS and argue in favor of considering it as a therapeutic intervention in PD and related neurodegenerative diseases.

## 1. Introduction

Parkinson’s disease (PD) is a progressive neurodegenerative disorder characterized primarily by the degeneration of dopaminergic (DA) neurons in the substantia nigra pars compacta (SNpc), which often correlates with the presence of α-synuclein-containing Lewy bodies [1]. This pathological process culminates in a discernible decline in dopamine levels in the striatal region and is closely associated with the onset of a variety of motor and non-motor symptoms that severely affect the lives of millions of people around the world [2]. Although the exact etiology of PD remains elusive, there is growing consensus that oxidative stress is one of the key factors in the development of the disease, including both idiopathic and genetic PD. This concept finds substantiation in the identification of oxidized lipids and proteins within the *post mortem* SNpc tissue of individuals affected by PD [3,4]. The neurotoxin 6-hydroxydopamine (6-OHDA), a dopamine analogue, is known to induce significant oxidative stress, damaging DA neurons [5]. This selective catecholaminergic neurotoxin was identified more than 60 years ago [6] and remains one of the most commonly used toxins to produce lesions of nigrostriatal dopaminergic neurons in rats [7]. 6-OHDA exerts its cytotoxic effects via several well-described mechanisms, i.e., the intracellular or extracellular auto-oxidation of 6-OHDA and the direct inhibition of mitochondrial respiratory chain complex I and IV [8]. These actions can generate reactive oxygen species (ROS), leading to cellular damage, oxidation of cellular macromolecules, mutations in mitochondrial DNA, and the initiation of apoptosis through mitochondrial pathways, indicated by the release of cytochrome c and other proteins involved in the apoptosis [9]. ROS also impairs the ubiquitin–proteasome system, causing an accumulation of defective proteins, a pathology mirrored in PD [10]. The vulnerability of dopaminergic neurons is further heightened by factors like ROS-generating enzymes, essential for dopamine synthesis, and elevated intracellular iron levels, facilitating oxidative reactions [11]. Against this background, understanding the role of oxidative stress is crucial for the development of effective PD treatments and prevention strategies. While recent years have seen advancements, existing therapies, which have centered on restoring dopamine, are proving insufficient to halt neuronal loss or mitigate the side effects associated with current treatments for PD. Repetitive transcranial magnetic stimulation (rTMS) is a form of non-invasive and painless brain stimulation that has shown therapeutic potential in many neurodegenerative disorders, including PD [12]. The exact mechanism of action remains elusive, but there is evidence that it can bring long-lasting benefits, that may last weeks or even months after the last stimulation [13]. rTMS can exert significant neuroprotective effects by acting on neuronal metabolism, neuroinflammation, and excitotoxicity, and by acting as a potent antioxidant and neuromodulator [14,15]. Intermittent theta burst stimulation (iTBS) is a specialized rTMS protocol that elicits an LTP-like increase in cortical excitability, and is proving to be an attractive and superior choice for neuromodulatory treatments in clinical disorders, mainly due to the rapid onset of modulatory effects compared to conventional rTMS [16]. In addition, our team’s research has demonstrated the marked efficacy of iTBS in attenuating inflammation and oxidative stress in various models of neurodegenerative disease [17,18,19,20,21]. However, the precise effects of iTBS on oxidative stress parameters in the context of PD remain largely unexplored. Therefore, the aim of the present study was to investigate the effects of iTBS protocol on the modulation of oxidative/antioxidative parameters and to determine the extent to which this modulation might be beneficial in a 6-OHDA-induced PD model.

## 2. Materials and Methods

### 2.1. Animals and Housing Conditions

A total of 16 two-month-old male Wistar rats (*n* = 16, 250 ± 30 g) housed at the Center of Veterinary Services animal facility, University of Defense, were used in this study. Animals (3–4/cage) were kept in constant environmental conditions (temperature of 23 ± 2 °C, a 12 h light–dark cycle) and ad libitum access to a standard diet and tap water. All animal care and experimental procedures in this study were performed in accordance with the “3Rs” principles and procedures as in the European Union Directive (2010/63/EU) and were approved by the Ethics Committee for Animal Experiments of the University of Belgrade—Faculty of Biology (No. 323-07-08250/2021-05).

### 2.2. Unilateral 6-Hydroxydopamine Lesion of the Right Substantia Nigra Pars Compacta

Before initiating the lesioning process, rats were sedated with a mixture of ketamine (100 mg/kg) and xylazine (10 mg/kg) to ensure their well-being and minimize discomfort, and then fixed in a stereotaxic instrument (Stoelting Co., Wood Dale, IL, USA). A dose of 2 µL of 6-OHDA (6 μg/μL, Catalog No. 28094-15-7) dissolved in sterile saline containing 0.2% ascorbic acid was administered into the right SNpc (rSNpc) area. Conversely, a similar volume of saline was injected into the left SNpc (lSNpc) as a control. Injection coordinates were set at −5.40 mm AP, ± 2.10 mm ML, and + 7.40 mm DV, as indicated in the stereotaxic atlas of Paxinos and Watson. Because the stereotaxic references were derived for Wistar rats weighing 290 g, we applied a correction to each stereotaxic coordinate for all animals weighing more or less than 290 g, as described in [22]. The neurotoxin was infused through a 50-µL Hamilton syringe at a constant flow rate of 0.4 µL/min (Microinjector; Harvard Apparatus, Holliston, MA, USA) and the needle remained in place for an additional 5 min post-injection to ensure proper diffusion within the SNpc and was then carefully withdrawn [23]. The anesthesia administered during the injection procedure would have been sufficient to relieve pain for a reasonable duration following surgery. Immediately after the surgery, each animal received subcutaneously 1 mL of sterile saline to hydrate them through the anesthesia. To further minimize discomfort to the animals, buprenorphine (0.05 mg/kg) was administered subcutaneously every 24 h for three days, and the animals were monitored daily by a veterinarian in charge. Following the 6-OHDA injections, the rats were divided into two groups for further investigation: those that received iTBS sham stimulation (iTBSsh; *n* = 7) and those that underwent actual iTBS treatment (iTBS; *n* = 7). These animals were exposed to the respective treatments for 21 days before being humanely euthanized by decapitation (Harvard Apparatus, Holliston, MA, USA).

### 2.3. Rotarod Performance Test

To evaluate the impact of surgical interventions on motor coordination and balance, a Rotarod test was conducted. To accustom the animals to the Rotarod apparatus, they were first given three training sessions. During these sessions, they were first placed on an immobile cylinder for 30 s to encourage them to avoid falling. Subsequently, they were then placed on a rotating cylinder for 90 s, which moved at a constant speed of 10 revolutions per minute (rpm). Animals were tested one day before the operation to obtain baseline values and one day before stimulation began to assess changes in motor coordination caused by surgery. Test performance before stimulation served as a criterion for animal selection, i.e., to exclude animals that did not exhibit motor dysfunction (*n* = 2 animals). Animals underwent three test sessions, each session included three trials with an acceleration of 4 to 20 rpm and a maximum duration of 200 s per trial and a 30-min interval between trials [23]. Latency to fall and distance traveled were recorded for each animal, and the best performance from the trial was used for analyses.

### 2.4. Theta Burst Stimulation Protocol

The iTBS was administered using a MagStim Rapid2 system equipped with a 25 mm figure-of-eight coil (MagStim Company, Whitland, UK). The protocol consisted of twenty sequences of ten bursts (each burst contained 3 pulses at 50 Hz) and a repetition rate of 5 Hz. There was a 10 s rest period between each sequence, culminating in a total stimulation time of 192 s per session. The intensity of the magnetic stimulation was maintained at 35% of the device’s maximum output, generating a magnetic field strength of 690 mT [17,23]. Animals were gently held during stimulation while they were allowed to move freely during the 10 s interval between restraints. The sham group (iTBSsh) was exposed to the noise artifact by placing the cage containing two animals next to the stimulation device while held gently to reproduce the mild restraint stress. All animals started the treatment 7 days after intoxication with the occurrence of the first motor symptoms. The same treatment protocol was repeated for 21 consecutive days. The iTBS protocol used in the present study is a whole-brain stimulation that affects the CPu and SNpc, among other brain regions, with E-field strengths above 28 V/m, which is sufficient to generate action potentials. Our previous research provides a 3D FEM model showing the geometry and gradient of magnetic and electric field density throughout the brain [23].

### 2.5. Blood Serum and Brain Tissue Collection 

After decapitation, whole blood was collected in anticoagulant-free containers. For serum analysis, the collected samples were allowed to coagulate for 30 min, followed by centrifugation at 1000× *g* for 10 min. The resulting supernatant was designated as serum. Multiple aliquots of serum were stored and frozen for subsequent analysis. In addition, following decapitation, the brains (*n* = 7/group) were swiftly extracted from the skull and rinsed with ice-cold saline. The right and left substantia nigra pars compacta and caudoputamen (rSNpc, lSNpc, rCPu, and lCPu) were dissected, frozen in liquid nitrogen, and stored at −80 °C. The samples were manually homogenized using a Teflon/glass homogenizer with 0.32 M sucrose in 5 mM Tris-HCl buffer at pH 7.4 (1 g wet tissue/10 mL buffer). The resulting homogenates were then centrifuged at 3000× *g* for 10 min at 4 °C, and the obtained supernatants were collected for further analysis [24]. Protein concentration was determined after isolation using the Pierce^TM^ BCA Protein Assay Kit (Cat. No. 23225; Thermo Scientific, Waltham, MA, USA), following the manufacturer’s instructions.

### 2.6. Malondialdehyde Determination

Reactive oxygen species can initiate a chain reaction in polyunsaturated lipids that leads to the formation of products such as malondialdehyde (MDA). To measure the MDA concentration, we followed the spectrophotometric method outlined by Girotti et al. [25]. In this method, samples were combined with a mixture of thiobarbituric acid (TBA) and Tris-HCl (pH 7.4) and then heated at 100 °C for 60 min. The reaction between MDA and TBA resulted in a red supernatant whose absorbance was measured spectrophotometrically at 535 nm. The variations in MDA levels were quantified as μmol of MDA per milligram of protein. The assay was performed in duplicate, and the mean values were reported along with the standard deviation.

### 2.7. Superoxide Anion Radical Determination

Superoxide anion radical (O_2_^•−^) quantification involved a spectrophotometric approach wherein nitro blue-tetrazolium (NBT; Merck, Darmstadt, Germany) reduction occurred in an alkaline, nitrogen-saturated medium [26]. The resulting yellow-colored reduced product, which was proportional to the superoxide radical concentration, was measured at 550 nm using an Ultrospec 2000 spectrophotometer. The results were expressed as nmol of reduced NBT per minute per milligram of protein.

### 2.8. Nitric Oxide Determination

Nitrosative stress was assessed by measuring nitrite and nitrate (NO_2_ + NO_3_) concentrations in the deproteinized samples. The combined concentration of NO_2_ + NO_3_ was determined using a spectrophotometric method at 492 nm. Nitrites were directly assayed using the Griess colorimetric method, which involved the use of 1.5% sulfanilamide in 1 M HCl and 0.15% N-(1-naphthyl) ethylenediamine dihydrochloride in distilled water. Conversely, nitrates were converted into nitrites through cadmium reduction prior to analysis [27,28]. The concentrations of nitrites in the samples were determined based on a standard curve generated using known nitrite concentrations and expressed as μmol/mg protein.

### 2.9. SOD Assay

The determination of the total superoxide dismutase activity (tSOD) was performed using a spectrophotometric approach based on the measurement of the decrease in the rate of spontaneous epinephrine auto-oxidation at 480 nm. The kinetic activity was monitored in a carbonate buffer, after the addition of 10 mM of epinephrine (Sigma, St. Louis, MO, USA) [29,30]. The obtained results were expressed as units per milligram of total protein (U/mg protein), where one unit represents the amount of enzyme required to inhibit epinephrine auto-oxidation by 50%.

### 2.10. Catalase Assay

The determination of catalase (CAT) activity involved a spectrophotometric method, where the formation of a yellow complex between ammonium molybdate (Serva, Feinbiochemica, Heidelberg, Germany) and H_2_O_2_ was monitored at 405 nm [31]. Data were expressed as mU of CAT per mg of protein. One unit of CAT activity is defined as µM H_2_O_2_/min/mg protein.

### 2.11. GSH Content Determination

The spectrophotometric assay for glutathione (GSH) utilizes the oxidation of GSH by the sulfhydryl reagent 5,5′-dithio-bis(2-nitrobenzoic acid) (DTNB), resulting in the formation of the yellow derivative 5′-thio-2-nitrobenzoic acid (TNBA), which is measurable at 412 nm. The formed glutathione disulfide (GSSG) could be regenerated to GSH by glutathione reductase in the presence of NADPH. The quantified results were expressed in nmol per mg of protein [32].

### 2.12. Total Sulfhydryl Groups (SH) Determination

The concentration of total sulfhydryl (SH) groups in tissue homogenates was measured spectrophotometrically at 412 nm in a phosphate buffer (0.2 mol + 2 mmol EDTA, pH 9) using 5,5-dithiobis-2-nitrobenzoic acid (DTNB, 0.01 M) [33]. The results were expressed as nanomoles of SH per milligram of protein.

### 2.13. Statistical Analyses

Normality of all data was assessed using the Shapiro–Wilk test, and the appropriate parametric or nonparametric tests were applied accordingly. Results of tests from serum were evaluated with an unpaired, two-tailed Student’s *t*-test, whereas tests from tissue samples were evaluated with a two-way ANOVA with treatment and hemisphere as the two factors. All ANOVA results are represented in Table 1, while the post hoc data are in the Section 3. We compared the left vs. right hemisphere in both the sham and iTBS group, and only right hemispheres between two groups. Values are presented as mean ± SD, as indicated in the figure captions. A significance level of *p* < 0.05 was considered statistically significant. Data analysis and graphical representation were performed using the GraphPad Prism 9.0 software package (San Diego, CA, USA).

## 3. Results

### 3.1. Behavioral Outcomes after Unilateral 6-OHDA Injection

When 6-OHDA is injected into the SNpc, it induces specific selective damage to dopaminergic neurons and a subsequent loss of terminals of dopaminergic neurons in their projection areas (CPu), leading to a reduction in dopamine production, similar to that observed in PD patients. After animals underwent unilateral injection of 6-OHDA into the right SNpc, the precision of the stereotaxic injection was confirmed by motor behavior and histological assessments (Figure 1). The animals showed marked impairment of motor skills. Because of the unilateral rSNpc lesion, the animals had difficulty using the contralateral left limbs. The animals also showed a significant reduction in latency to fall (Figure 1A, *t* = 7.917, d_f_ = 13, *p* < 0.0001) and distance traveled (Figure 1B, *t* = 6.604, d_f_ = 13, *p* < 0.0001) compared to their baseline performance in the rotarod test assessed one day before the lesion (Figure 1A,B).

### 3.2. Effects of Prolonged iTBS Treatment on Oxidative Balance in the Caudoputamen of 6-OHDA-Induced Model of PD

Studies involving 6-OHDA have shown that the primary cause of specific dopaminergic neuron damage in models of PD is cell death triggered by oxidative stress [34]. To determine the changes in the caudoputamen homogenates (CPu) of iTBSsh and iTBS animals three weeks after the start of the treatment, we performed measurements of oxidative stress markers and nonenzymatic/enzymatic components of antioxidative protection (Figure 2). When we compare the right CPu hemispheres of sham (iTBSsh) and iTBS animals, we observed a significant reduction in MDA levels (*p* < 0.0001) as well as O_2_^•−^ (*p* < 0.01) and NO levels (*p* < 0.001) (Figure 2A). No changes between left and right hemisphere within the group iTBSsh or iTBS were observed for all three parameters. The effects of iTBS on antioxidative capacity were evaluated through enzymatic (tSOD, CAT; Figure 2B) and nonenzymatic (GSH, SH^−^; Figure 2C) components of an antioxidative system. We observed an almost 2-fold increase in tSOD activity (*p* < 0.0001), in catalase activity (*p* < 0.0001) as well as in GSH (*p* < 0.05) and SH^−^ (*p* < 0.05) levels when we compared the right CPu hemispheres of sham and iTBS animals. Similarly, as for other parameters, no changes between the left and right hemisphere within the group iTBSsh or iTBS for all antioxidative parameters were observed. 

### 3.3. Effects of Prolonged iTBS Treatment on Oxidative Balance in the Substantia Nigra Pars Compacta of 6-OHDA-Induced Model of PD 

The same set of measurements were performed to analyze changes in the SNpc homogenates of iTBSsh and iTBS animals three weeks after we started the treatment (Figure 3). When we compared the right CPu hemispheres of sham (iTBSsh) and iTBS animals, we observed a significant reduction in pro-oxidant parameters—MDA levels (*p* < 0.001) as well as O_2_^•−^ (*p* < 0.0001) and NO levels (*p* < 0.01) (Figure 3A). 

There were no changes when we compared the left and right hemispheres within the groups for MDA and NO, but there was a slight but significant increase in O_2_^•−^ levels (*p* < 0.01) when we compared the left and right hemispheres of iTBSsh. The effects of iTBS on antioxidative capacity were evaluated through enzymatic (tSOD, CAT; Figure 3B) and nonenzymatic (GSH, SH^−^; Figure 3C) components of an antioxidative system and we observed a significant increase in tSOD (*p* < 0.001), CAT (*p* < 0.01) as well as in GSH (*p* < 0.01) and SH^−^ (*p* < 0.01) levels when we compared the right CPu hemispheres of sham (iTBSsh) and iTBS animals. There were no changes/interactions when we compared the left and right hemispheres within the group iTBSsh or iTBS for all antioxidative parameters except SH^−^ levels where we observed a slight, but significant decrease in the right hemisphere of iTBS animals when we compared it with left iTBS (*p* < 0.05). 

### 3.4. Effects of Prolonged iTBS Treatment on Oxidative Balance in the Serum of 6-OHDA-Induced Model of PD

We found that iTBS-treated animals had lower serum MDA levels than iTBSsh animals (Figure 4A; *t* = 4.264, d_f_ = 8; *p* = 0.0027). Another pro-oxidant parameter was also significantly reduced in iTBS-treated animals, NO (Figure 4B, *t* = 3.646, d_f_ = 8; *p* = 0.0065). Finally, we examined a non-enzymatic antioxidant parameter, SH, whose serum level was significantly increased in iTBS-treated animals (Figure 4C; *t* = 2.713, d_f_ = 8: *p* = 0.0265).

## 4. Discussion

Within this study, we evaluated the potential therapeutic benefits of the iTBS protocol at the molecular and systemic levels in the context of oxidative balance in a 6-OHDA-induced Parkinson’s disease model. The unilateral injection of 6-OHDA into the right SNpc is a widely recognized method for modeling PD. This approach leads to the targeted destruction of dopaminergic (DA) neurons and progressive degeneration of the nigrostriatal pathway [23]. Although 6-OHDA-induced degeneration does not mimic the best-known feature of PD, the formation of Lewy bodies, it does produce robust and relatively stable lesions without spontaneous recovery, effectively mimicking the primary behavioral and histopathological aspects of human PD [35]. In addition, once 6-OHDA enters the cell via the dopamine transporter, it is involved in auto-oxidation, intraneuronal generation of ROS, and ultimately apoptosis [36]. Many of these effects are thought to reflect processes in the PD brain, so the 6-OHDA model has a high degree of construct validity, making the model a perfect tool for studying different neuroprotective strategies. Some research indicates an increase in oxidative stress and its indicators in the brain and cerebrospinal fluid (CSF) of individuals with PD. Postmortem examinations revealed significantly increased levels of MDA, a by-product of lipid peroxidation, in the substantia nigra of PD sufferers—levels up to ten times higher than those in other brain regions and in individuals of the same age without PD [37]. Also, several studies have shown that injection of 6-OHDA into the SNpc causes a significant increase in MDA and TBARS levels [38,39]. Moreover, after injection of 6-OHDA into the striatum, one day after neurotoxin administration, an increase in the parameters of HNE, PC, and 3-NT, which are also products of oxidative damage, was observed, and this increase returned to baseline levels by the seventh day [40]. Several different studies using 6-OHDA have consistently shown a significant reduction in the activity of essential enzymes such as superoxide dismutase (SOD), catalase (CAT), and glutathione S-transferase (GST), which are critical for defense against the damaging effects of reactive oxygen species (ROS) in brain regions such as the striatum and substantia nigra [41,42,43,44,45]. Given the close link between oxidative stress and the etiology of PD, extensive efforts have been made to develop strategies aimed at restoring the balance between ROS/RNS and antioxidant mechanisms to mitigate the progression and severity of the disease. Due to its auto-oxidant properties, levodopa, the primary treatment for Parkinson’s disease, may exacerbate the disease progression rather than halting it by increasing oxidative stress and stimulating proinflammatory cytokine secretion, leading to neuroinflammation and subsequent dopaminergic neuron death [46]. In light of these challenges, our study explored the possibility of iTBS acting as an antioxidant, where an antioxidant could be considered in a broader sense as something that is able to stop the harmful effects of oxidative stress by either scavenging/decreasing ROS or stimulating the natural antioxidant system by targeting both neuronal and glial cells [47,48].

We demonstrated that after 21 days of iTBS stimulation, significant changes in antioxidant and pro-oxidant parameters occurred in the right hemisphere of the SNpc and CPu in animals subjected to iTBS treatment. Enzymatic antioxidants, including total superoxide dismutase (tSOD) and catalase (CAT), as well as non-enzymatic antioxidants such as reduced glutathione (GSH) and sulfhydryl groups (SH^−^) showed a significant increase. Conversely, the pro-oxidative markers malondialdehyde (MDA), nitric oxide (NO), and superoxide anion (O_2_^•−^) decreased. These changes were consistent in both the SNpc and CPu of iTBS animals. However, the absence of differences between left and right hemispheres within each group suggests that neurotoxic events associated with neurodegeneration and oxidative stress likely subside well before the 21-day sham/stimulation period ends. It has been shown that the peak of neuronal death is reached around the 7th day after intoxication, while oxidative stress occurs within the first week of use and then returns to baseline levels [49]. This highlights the need to interpret the results considering the broader effects of iTBS beyond localized brain regions. Accordingly, serum analysis revealed a decrease in MDA and NO levels together with an increase in the antioxidant defense parameter SH^−^, confirming the positive effects of iTBS and indicating its systemic effect. Building on our previous results showing recovery of motor performance and positive effects on emotional behavior, learning, and memory as well as decreased histopathological signs and neuroinflammation in a 6-OHDA-induced SNpc degeneration paradigm of PD [23], our present study extends the therapeutic benefits of iTBS. Furthermore, iTBS treatment in other neurodegeneration models, such as streptozotocin (STZ)-induced Alzheimer’s-like disease and trimethyltin (TMT)-induced Alzheimer’s-like disease, demonstrated a significant reduction in oxidative stress markers and increased antioxidant capacity [17,21]. One possible explanation for the observed improvement in the general oxidation status after iTBS lies in the nuclear transcription factor E2-related factor 2 (Nrf2), which is recognized as a central player in managing the oxidative stress response. Nrf2 associates with antioxidant response elements (ARE) upon nuclear entry, triggering the activation of genes related to this response pathway and promoting the expression of various antioxidant genes, thereby enhancing cellular defenses against oxidative damage [49]. The critical involvement of Nrf2-mediated signaling pathways in PD has been demonstrated by microchip analysis of different tissues from PD patients. This revealed a reduction in the expression of 31 genes with ARE sequences to which NRF2 binds [48]. The activation of the Nrf2 signaling pathway by rTMS has the potential to modulate the expression of antioxidant proteins like HO-1 and SOD1 [50]. This activation could reduce the damage caused by oxidative stress and protect brain tissue. Consistent with these findings, our previous studies have shown that Nrf2 is increased after iTBS treatment [21]. Furthermore, brain-derived neurotrophic factor (BDNF), which is known for its role as a modulator of synaptic plasticity, has been shown to induce the nuclear translocation of Nrf2. In our previous study, we found an increased protein expression of BDNF after 21 days of iTBS stimulation in 6-OHDA-induced PD [21,23]. Collectively, these factors could contribute to the observed enhancement in the general state of oxidative balance.

The considerable differences in the predefined settings of various rTMS protocols, such as the intensity of the device, the duration of the sessions, and the timing of application, represent a major hurdle to the standardization and comparison of the results of different studies. However, the iTBS paradigm proves to be a powerful excitatory protocol in the field of rTMS, characterized by consistent parameters as documented in the literature. In particular, it shows the same or even better efficacy at a shorter exposure time, making it a compelling approach for both human and animal studies. Nevertheless, there are some limitations to our study, one concerning a disadvantage of this technique due to the size and manual placement of the coil, which prevents focal stimulation of specific areas and categorizes the technique as a form of whole-brain stimulation. Consequently, the observed effects of iTBS may be due to a combination of cortical and subcortical stimulation, which is emphasized by their complex interconnectivity. The second limitation relates to the use of the 6-OHDA model. This model, although most commonly used, recapitulates only one main feature of the human pathology and the effects caused by this feature, namely dopamine deficiency. This model lacks the neuroinflammatory component and progressivity, meaning that it often produces changes at the synaptic level, i.e., synaptopathy, which is probably why we did not observe oxidative stress four weeks after intoxication. Finally, an important question and limitation that our manuscript does not address due to its primary objective is whether the observed effects are due to a primary pathology, i.e., 6-OHDA intoxication, or whether the same effects would also be observed in healthy, untreated animals, which requires further research. Nevertheless, the results obtained so far suggest that prolonged iTBS stimulation contributes to a better balance of ROS/RNS, possibly through its antioxidant properties. This effect is likely achieved by increasing the activity of enzymes that scavenge harmful oxidative molecules, thus protecting neurons from oxidative stress and contributing to their maintenance. In addition, iTBS may have overall systemic benefits that contribute to its therapeutic potential. Furthermore, it appears that iTBS has a cumulative effect and that the reduction in pro-oxidant parameters and the increase in antioxidant capacity is a result of whole brain stimulation and the effect persists even after 6-OHDA-induced degeneration and oxidative damage. The promising results of this study underscore the importance of further investigation of the specific signaling pathways and molecular cascades involved in the observed modulation of oxidative/antioxidant parameters. Future research efforts should focus on elucidating these precise mechanisms to solidify the understanding of how iTBS exerts its neuroprotective effects, thereby facilitating the development of targeted and effective therapeutic interventions for PD and related neurodegenerative diseases.

## 5. Conclusions

The study demonstrates that 21 days of iTBS treatment significantly bolsters antioxidative defenses in a rat model of PD induced by 6-OHDA, particularly in the critical brain regions SNpc and CPu. Additionally, serum analysis confirms iTBS’s systemic antioxidative impact, highlighting its potential in combating oxidative stress and neurodegeneration.

## Figures and Tables

**Figure 1 antioxidants-13-00218-f001:**
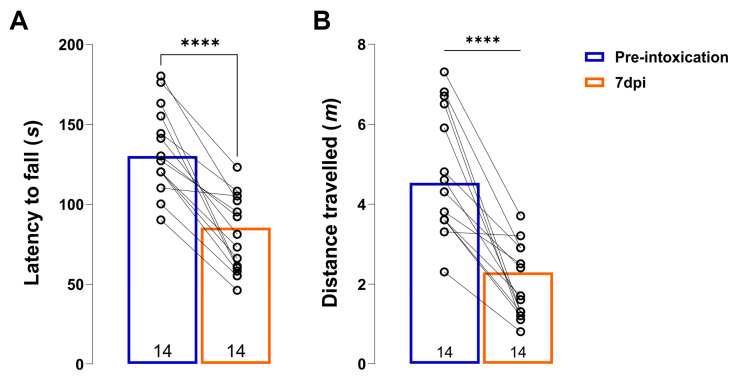
Unilateral 6-OHDA lesion induces motor impairment. Histograms showing latency to fall (**A**) and distance travelled (**B**) on a rotarod test of animals before (pre-intoxication) and 7-days after 6-OHDA unilateral lesion (7 dpi). All data are represented as mean ± SD. Dots in the graphs represent individual values. Number in the bottom of the graphs represents number of individual animals included in analysis. Results of post hoc Tukey’s test and significance are shown inside graphs: **** *p* < 0.0001.

**Figure 2 antioxidants-13-00218-f002:**
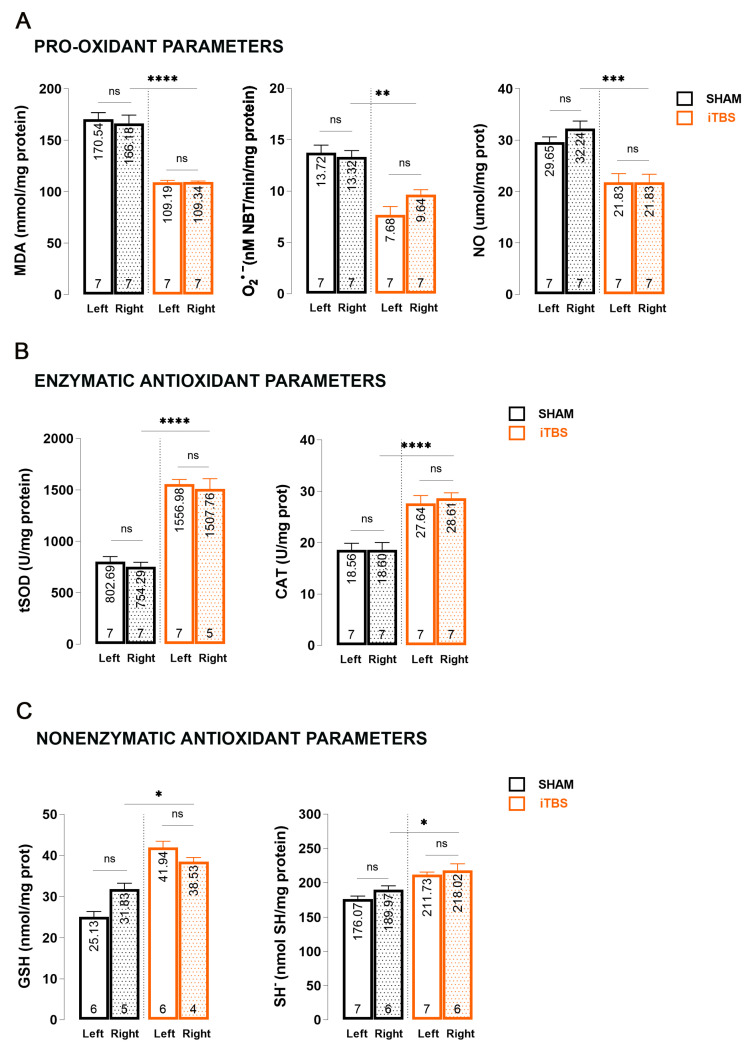
Effects of prolonged iTBS treatment on oxidative balance in the caudoputamen of 6-OHDA-induced model of PD. Spectrophotometric analysis of pro-oxidative and enzymatic/non-enzymatic antioxidative parameters: (**A**) MDA, O_2_^•−^, NO; (**B**) total SOD, CAT; (**C**) GSH and SH^−^ levels measured in caudoputamen homogenates (left and right hemisphere) from sham and iTBS animals after three weeks of stimulation. Bars shows mean activity expressed as U/mg protein or mol/mg protein. All data are presented as mean ± SD. Number in the bottom of the graphs represent number of individual animals included in analysis. Results of post hoc Tukey’s test and significance are shown inside graphs: ns–not significant, * *p* < 0.05, ** *p* < 0.01, *** *p* < 0.001, **** *p* < 0.0001.

**Figure 3 antioxidants-13-00218-f003:**
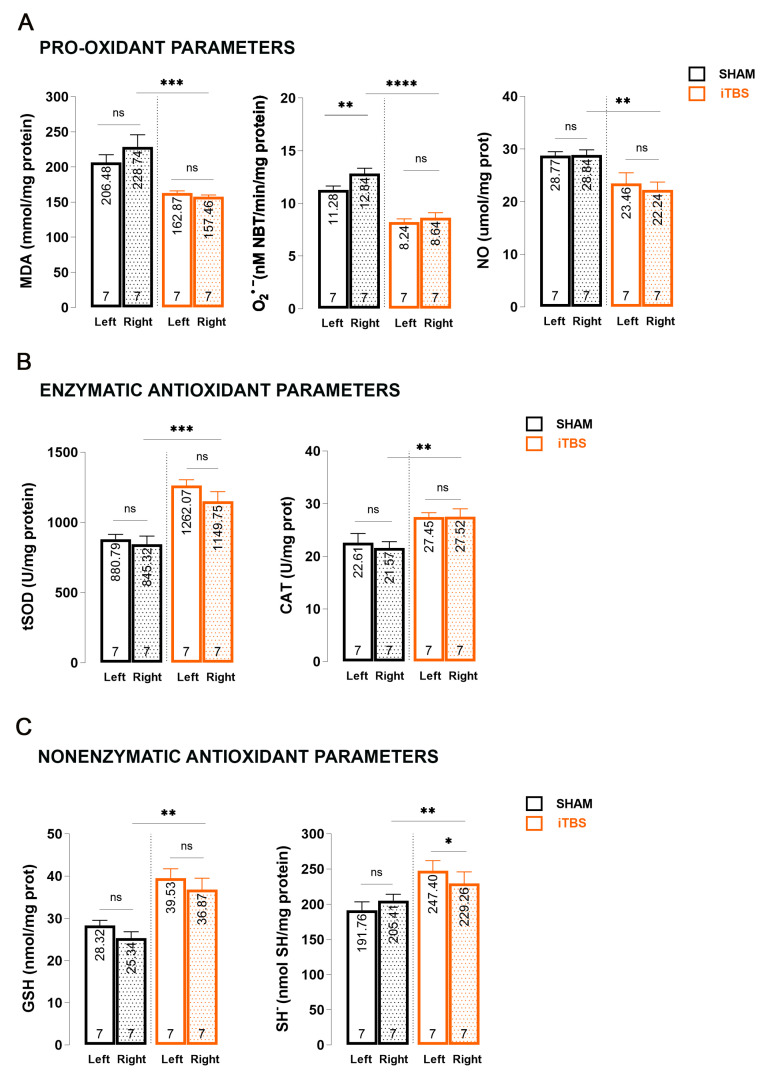
Effects of prolonged iTBS treatment on oxidative balance in the substantia nigra pars compacta of 6-OHDA-induced model of PD. Spectrophotometric analysis of pro-oxidative and enzymatic/non-enzymatic antioxidative parameters: (**A**) MDA, O_2_^•−^, NO; (**B**) total SOD, CAT, (**C**) GSH and SH^−^ levels measured in midbrain homogenates (left and right hemisphere) from sham and iTBS animals after three weeks of stimulation. Bars shows mean activity expressed as U/mg protein or mol/mg protein. All data are presented as mean ± SD. The numbers at the bottom of the graphs indicate the number of individual animals included in the analysis. Results of post hoc Tukey’s test and significance are shown inside graphs: ns—not significant, * *p* < 0.05, ** *p* < 0.01, *** *p* < 0.001, **** *p* < 0.0001.

**Figure 4 antioxidants-13-00218-f004:**
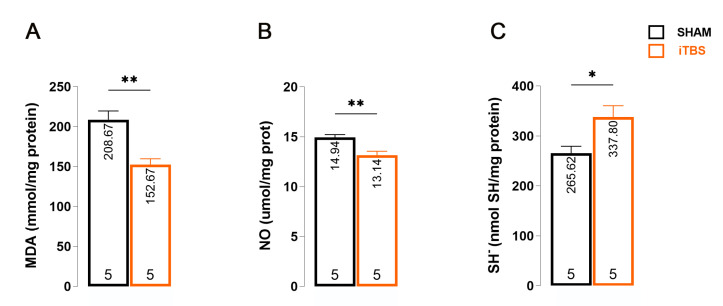
Effects of prolonged iTBS treatment on oxidative balance in the serum of 6-OHDA-induced model of PD. Spectrophotometric analysis of pro-oxidative and antioxidative parameters: (**A**) MDA-, (**B**) NO-, (**C**) SH^−^ levels measured in blood serum of sham and iTBS animals after three weeks of stimulation. The bars show the mean activity expressed as U/mg protein or mol/mg protein. All data are presented as mean ± SD. The numbers at the bottom of the graphs indicate the number of individual animals included in the analysis. Results of post hoc Tukey’s test and significance are shown inside graphs: * *p* < 0.05, ** *p* < 0.01.

**Table 1 antioxidants-13-00218-t001:** Results of the two-way ANOVA test.

Parameters	SNpc	CPu
MDA	Interaction: F_(1, 21)_ = 1.22; *p* = 0.2814	Interaction: F_(1, 22)_ = 3.44; *p* = 0.0769
	Hemisphere: F_(1, 21)_ = 2.13; *p* = 0.1593	Hemisphere: F_(1, 22)_ = 3.30; *p* = 0.0826
	Treatment: F_(1, 21)_ = 42.51; *p* < 0.0001	Treatment: F_(1, 22)_ = 211.0; *p* < 0.0001
O_2_^•−^	Interaction: F_(1, 20)_ = 4.02; *p* = 0.0585	Interaction: F_(1, 18)_ = 0.83; *p* = 0.3730
	Hemisphere: F_(1, 20)_ = 13.62; *p* = 0.0015	Hemisphere: F_(1, 18)_ = 0.83; *p* = 0.3729
	Treatment: F_(1, 20)_ = 83.32; *p* < 0.0001	Treatment: F_(1, 18)_ = 220.1; *p* < 0.0001
NO	Interaction: F_(1, 20)_ = 0.66; *p* = 0.4235	Interaction: F_(1, 24)_ = 0.79; *p* = 0.3805
	Hemisphere: F_(1, 20)_ = 0.003; *p* = 0.9563	Hemisphere: F_(1, 24)_ = 0.79; *p* = 0.3805
	Treatment: F_(1, 20)_ = 22.00; *p* = 0.0001	Treatment: F_(1, 24)_ = 39.77; *p* < 0.0001
tSOD	Interaction: F_(1, 21)_ = 0.66; *p* = 0.8945	Interaction: F_(1, 18)_ = 0.83; *p* = 0.3730
	Hemisphere: F_(1, 21)_ = 0.003; *p* = 0.1376	Hemisphere: F_(1, 18)_ = 0.83; *p* = 0.3729
	Treatment: F_(1, 21)_ = 22.00; *p* < 0.0001	Treatment: F_(1, 18)_ = 220.1; *p* < 0.0001
CAT	Interaction: F_(1, 20)_ = 2.27; *p* = 0.1471	Interaction: F_(1, 21)_ = 1.42; *p* = 0.2463
	Hemisphere: F_(1, 20)_ = 1.31; *p* = 0.2656	Hemisphere: F_(1, 21)_ = 0.68; *p* = 0.4158
	Treatment: F_(1, 20)_ = 22.32; *p* = 0.0001	Treatment: F_(1, 21)_ = 49.55; *p* < 0.0001
GSH	Interaction: F_(1, 20)_ = 0.21; *p* = 0.6476	Interaction: F_(1, 18)_ = 10.02; *p* = 0.0054
	Hemisphere: F_(1, 20)_ = 3.41; *p* = 0.0795	Hemisphere: F_(1, 18)_ = 0.04; *p* = 0.8333
	Treatment: F_(1, 20)_ = 35.64; *p* = 0.0001	Treatment: F_(1, 18)_ = 60.78; *p* < 0.0001
SH^−^	Interaction: F_(1, 21)_ = 5.80; *p* = 0.0252	Interaction: F_(1, 22)_ = 0.39; *p* = 0.5342
	Hemisphere: F_(1, 21)_ = 7.35; *p* = 0.0130	Hemisphere: F_(1, 22)_ = 2.80; *p* = 0.1081
	Treatment: F_(1, 21)_ = 13.32; *p =* 0.0015	Treatment: F_(1, 22)_ = 27.97; *p* < 0.0001

## Data Availability

The data presented in this study are available on request from the corresponding author. The data are not publicly available due to the policy of our Institute.

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
