# Peer review of "Sustained Systemic Antioxidative Effects of Intermittent Theta Burst Stimulation beyond Neurodegeneration: Implications in Therapy in 6-Hydroxydopamine Model of Parkinson’s Disease"

_antioxidants, 2024, doi:10.3390/antiox13020218_

Round 1

Reviewer 1 Report

Comments and Suggestions for Authors

This manuscript is very interesting as it aims to study the neuroprotective potential of intermittent theta burst stimulation 17 (iTBS) in 6-hydroxydopamine-induced PD model in rats. The authors conclude that "treatment with iTBS resulted in a significant decrease in oxidative markers, simultaneously increasing antioxidative parameters in the caudoputamen and substantia nigra pars clava well after 6-OHDA-induced neurodegeneration process was over. Serum analysis confirmed the systemic effect of iTBS with a decrease in oxidative markers and an increase in antioxidants. Prolonged iTBS exerts a modulatory effect on oxidative/antioxidant parameters in the 6-OHDA-induced PD model, suggesting a potential neuroprotective benefit, even though at this specific time point 6-OHDA-induced oxidative status was unaltered" The authors suggest that iTBS can be considered in a therapeutic intervention in PD.

These results are interesting and valuable from an experimental point of view but cannot be extrapolated to have a potentially therapeutic value in Parkinson's disease.

These studies were carried out in an experimental Parkinson's model based on 6-OHDA, whose value for translation to clinical studies is highly questioned. There is a long list of clinical studies based on exogenous neurotoxins such as 6-OHDA and MPTP that have failed even though preclinical studies with these neurotoxins were successful. It is a mistake to continue extrapolating the success in studies with exogenous neurotoxins such as 6-OHDA for their therapeutic application in Parkinson's disease. (Nat Rev Neurol. 2015 Jan;11(1):25-40. doi: 10.1038/nrneurol.2014.226. Epub 2014 Dec 2.)

Author Response

We would like to thank the reviewer for carefully reading our manuscript and for the comments that critically shaped crucial aspects of our work. We would also like to thank the reviewer for initiating a fruitful scholarly discussion that has improved this manuscript but will also influence future manuscripts.

We would like to mention that we have changed the manuscript by implementing the reviewer's comments as well as editorial suggestions.

We have addressed them point by point. The reviewer's comments are in bold, while our responses are in plain text.

These studies were carried out in an experimental Parkinson's model based on 6-OHDA, whose value for translation to clinical studies is highly questioned. There is a long list of clinical studies based on exogenous neurotoxins such as 6-OHDA and MPTP that have failed even though preclinical studies with these neurotoxins were successful. It is a mistake to continue extrapolating the success in studies with exogenous neurotoxins such as 6-OHDA for their therapeutic application in Parkinson's disease. (Nat Rev Neurol. 2015 Jan;11(1):25-40. doi: 10.1038/nrneurol.2014.226. Epub 2014 Dec 2.)

Thank you for your comment. We agree that the transferability of experimental data is often questioned when it comes to real clinical trials, but this should not underscore the importance of the experimental, preclinical part of the studies. If a drug or compound has shown a positive effect in a preclinical model and subsequently fails in a clinical trial, which is often the case, we believe this could be due to the difference between humans and rodents and the fact that no model fully represents the complexity of the disease. However, the 6-OHDA model has led to at least one FDA-approved drug that is now being used in clinical practice (please see PMID: 24687255). This suggests that the model itself is neither ``bad`` nor necessarily ``good``, but that it is sometimes important to target the aspect of the disease that is reproduced in the model. 6-OHDA is basically a synaptopathy, because there is rapid progression with transient and low-grade inflammation (please see), accompanied by degeneration of dopaminergic neurons triggered by the 6-OHDA compound mainly through oxidative stress (please see PMID: 37096171, PMID: 28466266). Oxidative stress accompanies virtually every degenerative process and neuroinflamamtion, thus we examined wheter rTMS might have a beneficial effects on this aspect of the disease. Givent that we have started rTMS when most of the neurons have already degenerated, rTMS modulation answers to a question what happens after prolonged exposure on parameters of both prooxidative and antioxidative status. Furthermore, we have shown an interesting systemic effect of rTMS in serum of these animals. Finally, rTMS has already been used in many clinical trials on PD patients and have shown generally promising results (please see PMID: 18703005). Given that rTMS is noninvasive tool, we belive that our study may spark other experimental studies but also some clinical trails which may examine some of these parameters. We have clearly stated limitations of our study in the disscusion section.

Reviewer 2 Report

Comments and Suggestions for Authors

This study is well organized and evaluates the therapeutic potency of iTBS in the ipsilateral 6-OHDA-injected rat model of Parkinson’s disease. The author showed that iTBS treatment mitigates the production of pro-oxidant factors and recovers antioxidant molecules in SNpc and CPu, and blood from 6-OHDA-injected rat. Although iTBS treatment may alleviate the dopaminergic neurodegeneration of PD via regulating ROS levels, the mechanisms of iTBS should be further investigated. This reviewer suggests few comments prior its publication.

Ipsilateral lesion leads to asymmetric movements, so many studies have conducted amphetamine-induced rotation tests to assess animal motor symptoms. In addition to rotarod test, add more behavior tests to evaluate the therapeutic potential of iTBS treatment. More importantly, the author should include the mice groups; 1) Vehicle-Sharm, 2) Vehicle-iTBS, 3) 6-OHDA-Sharm, and 4) 6-OHDA-iTBS.

Even though the author validates DA neurodegeneration from 6-OHDA-induced rat in previous works, please present the data regarding the number of TH-positive neurons in SNpc to confirm the correlation between ROS suppression and DA degeneration.

In the Fig3, as a therapeutic strategy, the antioxidative effect of iTBS may be more effective early onset of PD, before onset of DA neurodegeneration. It would be better to test the extent to which the antioxidant molecules produced by iTBS could prevent neurodegeneration in a period of less than 3 weeks.

In the Materials and Methods section, please provide information where the brain region is highly affected by iTBS treatment.

Finally, the author should present the result about the levels of pro- and anti-oxidant molecule in WT rat by iTBS treatment.

Author Response

We would like to thank the reviewer for careful reading of our manuscript and for comments, which critically shaped crucial aspects of our work and improved its quality.

We have responded point by point to all concerns. The reviewers' comments are in bold, while our responses are in plain text.

Ipsilateral lesion leads to asymmetric movements, so many studies have conducted amphetamine-induced rotation tests to assess animal motor symptoms. In addition to rotarod test, add more behavior tests to evaluate the therapeutic potential of iTBS treatment. More importantly, the author should include the mice groups; 1) Vehicle-Sharm, 2) Vehicle-iTBS, 3) 6-OHDA-Sharm, and 4) 6-OHDA-iTBS.

Thank you for your comment. We have analyzed the behavioral phenotype of 6-OHDA animals in detail in our previous study, including rotarod test, cylinder test, open field, sucrose preference, and novel object recognition test (see PMID: 37296646). The tissues from that experiment were also used in this study. In addition, we had a control group, a 6-OHDA group, a 6-OHDA sham group, and a 6-OHDA iTBS group in that experiment. Our main goal was to compare what happens in the 6-OHDA sham group and the 6-OHDA iTBS group in terms of oxidative stress, hence only two experimental groups.

Even though the author validates DA neurodegeneration from 6-OHDA-induced rat in previous works, please present the data regarding the number of TH-positive neurons in SNpc to confirm the correlation between ROS suppression and DA degeneration.

Тhank you for your comment. Since the tissues and samples are from the same experiment from which we published changes in TH expression (please see PMID: 37296646), we believe it would not be fair to repeat the same result twice. However, it would be interesting to count the cells in our future studies, as we cannot tell whether surviving neurons increase TH expression or whether fewer TH-positive neurons have degenerated based on the changes in TH expression we observed with the Western blot.

In the Fig3, as a therapeutic strategy, the antioxidative effect of iTBS may be more effective early onset of PD, before onset of DA neurodegeneration. It would be better to test the extent to which the antioxidant molecules produced by iTBS could prevent neurodegeneration in a period of less than 3 weeks.

Thank you for your comment. We agree that it would be a nice addition to examine the changes in oxidative/antioxidative status, however, when we started our study we relied on the existing literature, thus, taking these timepoints.

In the Materials and Methods section, please provide information where the brain region is highly affected by iTBS treatment.

Thank you for your suggestion. We have added that.

Finally, the author should present the result about the levels of pro- and anti-oxidant molecule in WT rat by iTBS treatment.

Thank you for your suggestion. Unfortunately, we are not in a position to conduct these experiments at the moment, but we will certainly do so in the future as we are conducting a study supported by a grant in which we are investigating the effects of iTBS on healthy Wistar rats.

Round 2

Reviewer 1 Report

Comments and Suggestions for Authors

No further comments.

Author Response

Thank you

Reviewer 2 Report

Comments and Suggestions for Authors

The authors already conducted several experiments suggested by this reviewer, and I also agree author’s response that there is no necessary to double confirm using same mice. However, to investigate the therapeutic role of iTBS in the brain, WT-iTBS groups are need to evaluate whether author’s finding mainly works in disease condition (specific to PD) or not.

Author Response

Thank you very much for your comments and for supporting the scientific discourse. Your insights and comments have significantly improved our manuscript. Our original aim of the study was to see if iTBS also affects oxidative stress in our experimental model. The main reason for this study is that we hypothesize that iTBS acts globally on the brain and is not selective for NMDA-mediated or purinergic signaling (our paper is currently under revision), but affects many signaling systems. Therefore, we wanted to characterize as many aspects and iTBS effects on them, that are known to contribute to PD and that are reproduced in the 6-OHDA model. Nevertheless, it is a very interesting question raised by Reviewer whether the effects we observed are due to 6-OHDA intoxication or whether these effects would also be observed in healthy, unprimed rats. It is well-known that rTMS generally, as a therapeutical approach has antioxidative effects in many experimental models of neurodegenerative disorders, mostly on AD models and ischemic models (please see PMID: 29385851). Even though, we are currently not able to provide what Reviewer has asked, we are submitting the data on our current research showing that iTBS effects are probably dependent on duration of stimulation and daily application. Data we are supplementing are from caudoputamen of healthy animals stimulated for five days, once per day, as opposed to our study where all groups were stimulated for three weeks, twice a day (we have examined also, other structures including hippocampus, prefrontal cortex and cerebellum).

Furthermore, in our previous study on cerebellar tissue from healthy rats, where animals were stimulated with iTBS once a day for ten days and only once (in a single session), we obtained different results for some parameters, suggesting that the bioneurochemistry behind iTBS is complex and that region and substance should also be considered as factors (please see PMID: 27623095). Specifically for this reason, we are conducting the current study to test the acute (single session), sub-chronic (7 days of stimulation twice daily) and early (7 days after the last session) and late after-effects (14 days after the last session) of iTBS in healthy animals to reveal the complex neurochemistry in an intact system.

Given that our initial goal of the study was just to compare these two groups, but that we acknowledge the importance of question raised by Reviewer, we are adding this in our Study limitation as a serious open question which need a further exploration and clarification.

Round 3

Reviewer 2 Report

This study shows the antioxidant property of iTBS treatment against 6-OHDA intoxicated rats. The data supports the therapeutic potential of iTBS treatment and adds valuable insights to our understanding of the molecular mechanisms in iTBS treatment. This reviewer recommends that the current version of the manuscript is acceptable for publication

Although the authors are hard to conduct the experiments with groups suggested by this reviewer's issue, they showed antioxidant properties of iTBS-treated WT animals with variable conditions. Moreover, they added a paragraph regarding limitation of this study in discussion section.